# Simultaneous two-photon imaging of action potentials and subthreshold inputs in vivo

Yuki Bando [1,2✉], Michael Wenzel [1,3] & Rafael Yuste [1]

To better understand the input-output computations of neuronal populations, we developed ArcLight-ST, a genetically-encoded voltage indicator, to specifically measure subthreshold membrane potentials. We combined two-photon imaging of voltage and calcium, and successfully discriminated subthreshold inputs and spikes with cellular resolution in vivo. We demonstrate the utility of the method by mapping epileptic seizures progression through cortical circuits, revealing divergent sub- and suprathreshold dynamics within compartmentalized epileptic micronetworks. Two-photon, two-color imaging of calcium and voltage enables mapping of inputs and outputs in neuronal populations in living animals.

[1] NeuroTechnology Center, Department of Biological Sciences, Columbia University, New York, NY 10027, USA. [2] Present address: Department of Organ and Tissue Anatomy, Hamamatsu University School of Medicine, Hamamatsu 431-3192, Japan. [3] Present address: Department of Epileptology, University of Bonn, 53127 Bonn, Germany. ✉email: bando@hama-med.ac.jp

Neurons receive synaptic inputs and generate action potential for their targets in complex neural networks. Thus, to understand the neural basis of behavior, it appears necessary to analyze the input and output relationship in neuronal populations in behaving animals. Currently, whole-cell patch-clamp, intra-, or extracellular recording, or calcium imaging are widely used to record neural activity. Input and output can be recorded simultaneously from only a few neurons at the same time, using whole-cell or intracellular recording in vivo[1,2], and it is impossible to record them over days. On the other hand, calcium imaging or multi-unit recording enable long-term recording of action potentials from multiple cells in vivo, but not of subthreshold inputs. A method to simultaneously record input and outputs of sizeable neuronal populations in vivo would be desired, yet appears challenging.

Optical measurements of voltage appear like the ideal solution to this challenge. Recently, the performance of genetically enco-ded voltage indicators (GEVIs) has improved[3–9]. Unlike organic voltage-sensitive dyes, genetically encoded sensors enable cell-type-specific labeling and cellular imaging. But for in vivo imaging with cellular resolution, two-photon voltage imaging is necessary. Using GEVIs, spatiotemporal mapping of spiking activity has been achieved in acute brain slices or in vivo[5–8,10]. However, simultaneous detection of subthreshold inputs and action potentials remains challenging. Previously, we tested existing GEVIs for two-photon imaging in vivo and found that ArcLight-MT could be used successfully to measure subthreshold membrane potentials, action potentials, and local field potentials (LFP), but that its signal-to-noise ratio (SNR) was still poor[11]. This is partly because existing GEVIs, including ArcLight-MT, are designed for the detection of action potentials, resulting in insufficient sensitivity to subthreshold potentials. As solution to this challenge, we explored the use of two indicators, one optimized for subthreshold potentials and another one for spikes. To pursue this, we modified ArcLight-MT to specifically improve its SNR to detect subthreshold membrane potential changes, by introducing a point mutation to negatively shift its voltage-dependency[12] (ArcLight-ST). Then, we combined ArcLight-ST with a red calcium indicator, jRGECO1a[13], and used two-photon excitation to measure action potentials and subthreshold membrane potentials with cellular resolution in vivo. Furthermore, two-photon, two-color imaging of voltage and calcium revealed differential network-level dynamics of subthreshold and spiking events during epileptic seizure pro-gression. The combination of voltage and calcium imaging could be useful to large-scale analysis of neural circuit dynamics in behaving animals.

## Results

**Modification of ArcLight-MT to increase sensitivity to sub-threshold potentials**. To improve the sensitivity of ArcLight-MT for subthreshold synaptic inputs, we modified ArcLight-MT and characterized the variants' properties. A previous report showed that replacing the arginine at position 217 with glutamine or glutamate in the *Ciona intestinalis* voltage-sensing domain (Ci-VSD) negatively shifted its voltage-dependency[12]. Based on this, we introduced a Q217E mutation in ArcLight-MT (named ArcLight-subthreshold or ST) to adjust its voltage-dependency. We also added the C-terminal region of the voltage-gated potassium channel, $K_v2.1$ to ArcLight-ST (Kv-ArcLight-ST) to localize ArcLight-ST to the soma, as this improves GEVI expression in vivo[5,8] (Fig. 1a). The C-terminal $K_v2.1$ peptide contributed to the somatic localization of ArcLight-ST (Supple-mentary Fig. 1), but did not affect the electrical properties of

neurons (Supplementary Fig. 2). We first expressed three Arc-Light variants in cultured hippocampal neurons under a CMV promoter and performed one-photon imaging with a sCMOS camera at 1 kHz. Excitation light was filtered at 480/40 nm, and excitation power was ~5.5 mW (~36 mW/mm$^2$) at the stage. As expected, ArcLight-ST and Kv-ArcLight-ST showed a more negative voltage-dependency than ArcLight-MT (Fig. 1c, d). We then tested the performance of these ArcLight variants during small depolarizing voltage steps from −70 mV, finding that ArcLight-ST and Kv-ArcLight-ST displayed larger optical signals than the original ArcLight-MT (Fig. 1e, f). This indicated that ArcLight-ST and Kv-ArcLight-ST are more sensitive to sub-threshold potentials than ArcLight-MT. As expected from the negative shift in voltage dependency, ArcLight-ST and Kv-ArcLight-ST showed lesser sensitivity than ArcLight-MT for spike detection, underscoring that ArcLight-ST and Kv-ArcLight-ST are specialized GEVIs for subthreshold events (Supplementary Fig. 3). All ArcLight variants showed similar photobleaching kinetics with both one-photon (imaging speed: 10 Hz, and exci-tation power: ~5.5 mW (~36 mW/mm$^2$) at the stage) and two-photon imaging at 940 nm (imaging speed: 30 Hz, and excitation power: ~28 mW (average power: 0.11 µW/pixel)) (Supplementary Fig. 4).

**Characterization of ArcLight variants in vivo**. Next, we tested these ArcLight variants in vivo with both one-photon wide-field and two-photon imaging in mice anaesthetized with isoflurane (~1.5% v/v) (Figs. 2–4). Imaging speed was 30 Hz in both one- and two-photon imaging. ArcLight variants were expressed by in utero electroporation in neocortical layer 2/3 pyramidal neurons under the CAG promoter. For one-photon wide-field imaging, excitation light was filtered at 480/40 nm, and excitation power was ~1 mW (~0.05 mW/mm$^2$) at the stage. For two-photon imaging, the excitation laser was tuned to 940 nm, and its power was 130–160 mW (0.50–0.61 µW/pixel) at the stage. Two-photon imaging was performed typically at 150–300 µm below the cor-tical surface. ArcLight-ST and Kv-ArcLight-ST showed better sensitivity than ArcLight-MT to visually evoked potentials with both one-photon wide-field and two-photon imaging in vivo. ArcLight-ST showed better SNR in one-photon wide-field ima-ging and two-photon imaging of dendritic compartments and optical field potential (OFP), an equivalent of electrical local field potential[11], whereas Kv-ArcLight-ST was most suitable for two-photon imaging with cellular resolution (Figs. 2–4, Supplemen-tary Fig. 7). These results suggest that the somatic localization of Kv-ArcLight-ST is important for single-cell resolution imaging, while the distribution of ArcLight-ST in all compartments including dendrites and axons is more suitable for imaging of field potentials and subcellular compartments. Since the fast component of the rising and decay time of ArcLight-ST and Kv-ArcLight-ST was ~60 ms, the acquisition frame rate of 30 Hz was close to the Nyquist frequency. The mean and standard devia-tion (S.D.) of $\Delta F/F$ around 0 mV ($|\Delta V_m| < 1$ mV) was −0.28 and 0.88, respectively. Thus, the detection limit of voltage change was calculated as 15.3 mV (= (mean($|\Delta V_m| <1$ mV) −3 × S.D.($|\Delta V_m| <1$ mV))/slope). Because of these properties of ArcLight-ST and Kv-ArcLight-ST, both sensors can only detect large and slow fluctuations of membrane potential. The imaging system noise was not significant in our imaging conditions in vivo (Supple-mentary Fig. 5). In one-photon wide-field imaging, small fluc-tuations of optical signals of ~7 Hz (= 420 peaks/min) were observed (Supplementary Fig. 5f). Since the frequency of this small baseline fluctuation was similar to the typical heart rate in mice (400–800 beats/min), they most likely reflected the heart

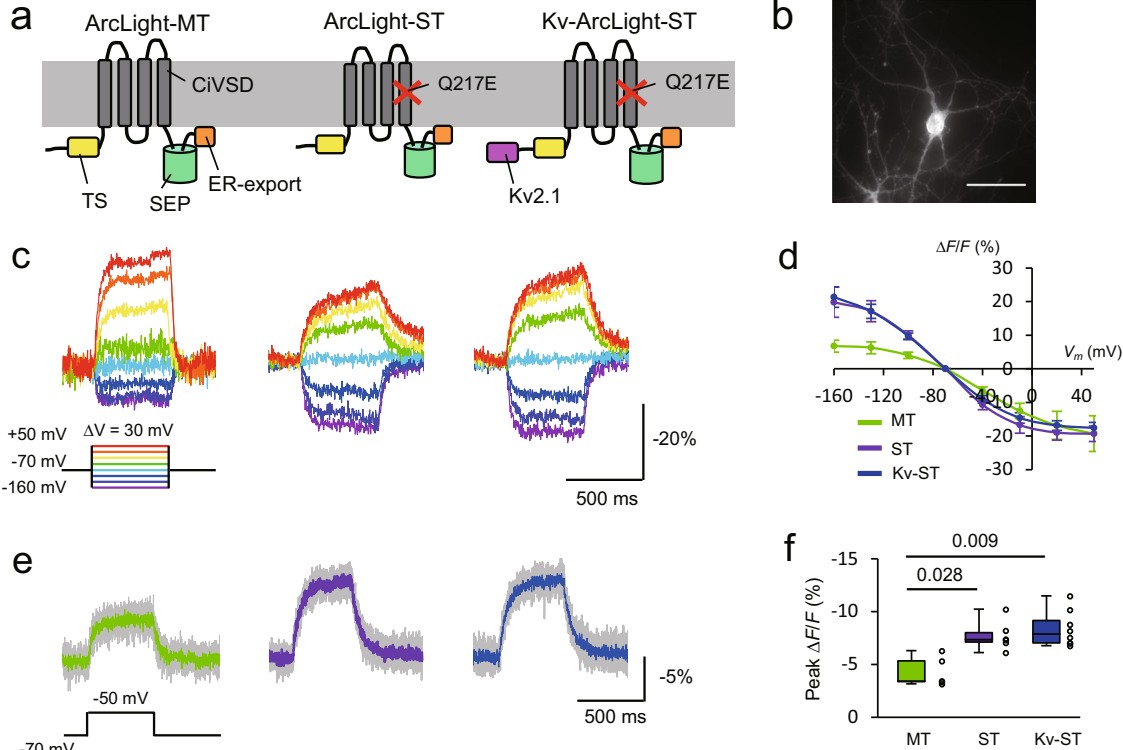

**Fig. 1 Design and characterization of ArcLight variants with one-photon imaging in vitro. a** Schematic drawing of ArcLight variants. TS, the export signal from the Golgi apparatus. SEP super ecliptic pHluorin. **b** A one-photon image of a cultured hippocampal neuron expressing Kv-ArcLight-ST. Scale bar, 50 μm. ArcLight variants were expressed under CMV promoter using calcium phosphate. **c** Optical signals during depolarizing and hyperpolarizing voltage steps in voltage-clamp mode. **d** Relationship between membrane potential and normalized fluorescence (mean ± SEM). $n = 5$ cells for each ArcLight variant. **e** Optical responses to small depolarizing voltage steps of 20 mV from −70 mV. Gray traces are individual trials, and colored traces are an average of five trials. **f** Peak fluorescence change in response to depolarizing voltage steps of 20 mV. $n = 5$ cells (ArcLight-MT), 6 (ArcLight-ST), 8 (Kv-ArcLight-ST). The Steel-Dwass test was used for statistical analysis. Excitation: 460–500 nm, ~5.5 mW (~ 36 mW/mm²) at the stage. Imaging was performed at 1 kHz. On each box, the central line represents the median, and the bottom and the top edges of the box represent the 25th and 75th percentile, respectively. The lower and the upper whiskers extend to the minimum and the maximum data points, respectively. Numbers on the graphs represent $P$ values.

beat. Two-photon imaging of Super Ecliptic pHluorin (SEP) A227D, a control indicator without a voltage-sensitive domain, did not show significant changes of fluorescence in response to visual stimulation, suggesting that the recorded optical signals using ArcLight variants were not affected by hemodynamic signals or change in cytosolic pH (Supplementary Fig. 6). As expected, the slow kinetics of ArcLight-ST and Kv-ArcLight-ST, although with an enhanced SNR, limit their temporal resolution. Indeed, although ArcLight-ST and Kv-ArcLight-ST showed larger SNR for subthreshold potentials, they did not perfectly trace voltage dynamics (Fig. 1e, Supplementary Fig. 8b) due to their slow kinetics. Nevertheless, in side to side comparisons, ArcLight-ST showed better performance than ASAP3 for detection of subthreshold activity in vivo, although ASAP3 can faithfully detect action potentials in vivo[8,10] (Supplementary Fig. 7). Furthermore, ArcLight-ST showed better performance than ASAP3 for the detection of dendritic signals (Supplementary Fig. 7). Using ArcLight-ST and Kv-ArcLight-ST measurements, we analyzed the spatiotemporal structure of large and slow subthreshold events in vivo. Discriminable patterns of activity were observed when wide fields of view (FOV) was used, but large and slow subthreshold activity was nearly synchronized in local cortical circuits (Figs. 3j–n, 4f–i). We conclude from these measurements and analyses that ArcLight-ST is effective for one-photon wide-field imaging and two-photon imaging from multiple subcellular compartments, while Kv-ArcLight-ST is most suitable for multi-

cellular two-photon population imaging of subthreshold potentials in vivo.

**Simultaneous two-photon voltage and calcium imaging in vivo.** To simultaneously record subthreshold inputs and action potentials, we then combined Kv-ArcLight-ST voltage imaging with calcium imaging. Previously, simultaneous voltage and calcium⁺ imaging in vivo have solely been employed using one-photon wide-field imaging, or two-photon line-scans of single cells[14,15], but our goal was to perform population imaging of both subthreshold and action potentials with cellular resolution. For simultaneous two-photon, two-color voltage and calcium imaging, we used a single excitation wavelength at 990 nm (150–180 mW at the stage (0.57–0.69 μW/pixel)) to excite both chromophores simultaneously. We performed two-photon imaging typically at a depth of 50–100 μm in brain slices, and 150–300 μm from the pial surface in vivo, at a 30 Hz acquisition rate, respectively. First, we explored whether the voltage and calcium imaging was possible with the same excitation wavelength and found that Kv-ArcLight-ST showed comparable performance at 990 nm vs. 940 nm. The red calcium indicator jRGECO1a also showed spike number-dependent fluorescence change at 990 nm (Supplementary Figs. 8, 9). Next, we tested whether the two-photon, two-color voltage, and calcium imaging could simultaneously discriminate subthreshold and spiking events in

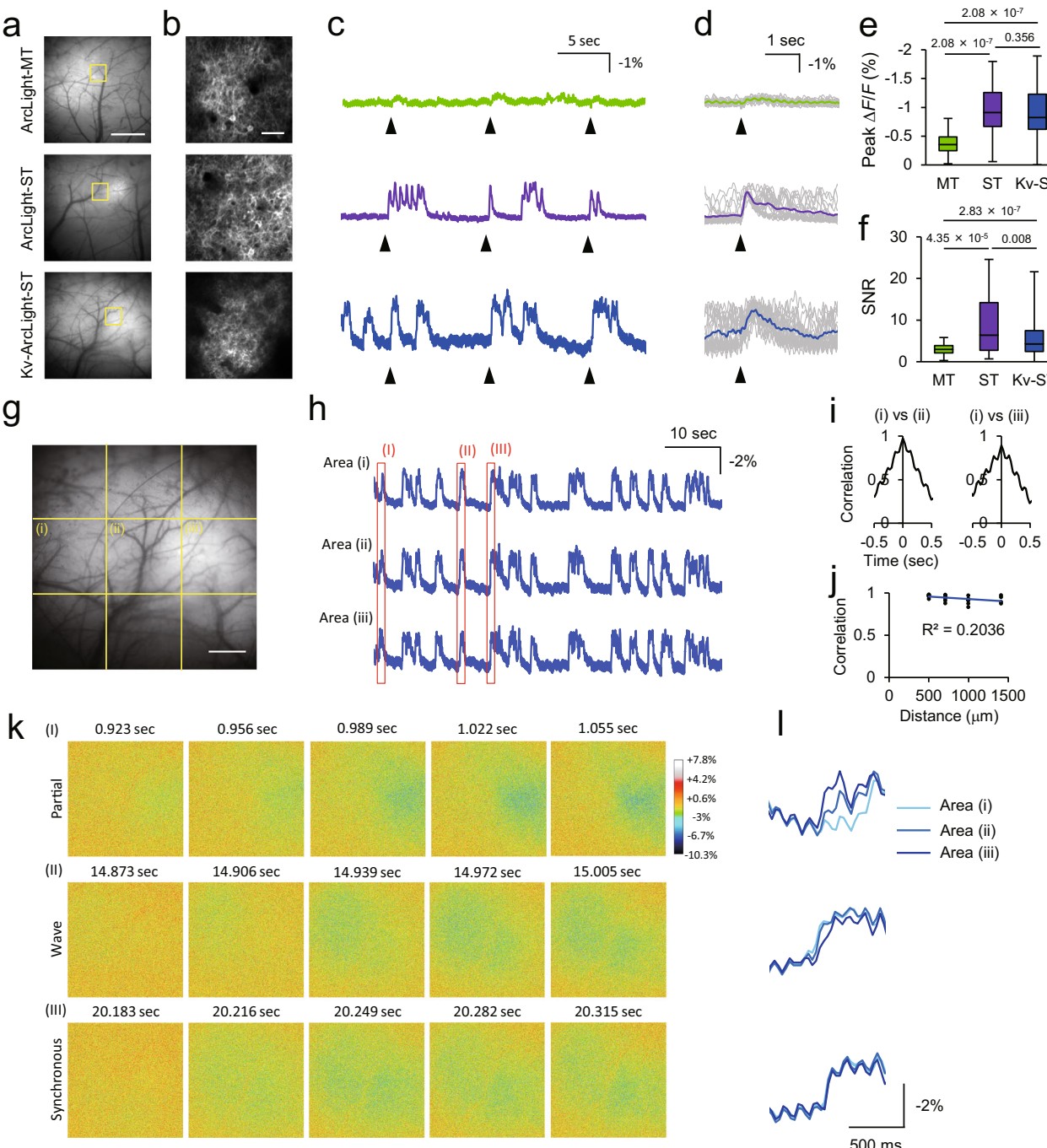

**Fig. 2 Performance of ArcLight variants with one-photon wide-field imaging in vivo. a** One-photon images of mouse primary visual cortex (V1) expressing ArcLight variants. Scale bar, 1 mm. **b** Two-photon images of areas indicated with yellow square in (**a**). Scale bar, 50 µm. **c** Representative optical traces of ArcLight variants. Arrowheads indicate the timing of visual stimulation with 10 ms flash light. **d** Stimulus-triggered average. Gray traces are individual trials, and colored traces are average over 20 trials. Arrowheads indicate the timing of visual stimulation. **e, f** Peak fluorescence change (**e**) and signal-to-noise ratio (SNR) (**f**) for visually evoked optical signals. SNR was calculated as peak $\Delta F/F$ over standard deviation of the baseline fluctuation for 0.5 s before visual stimulation. $n = 5$ FOVs from 5 mice in each condition. Steel-Dwass test was used for statistical analysis. **g–l** Spatiotemporal structure of subthreshold events in vivo. **g** An enlarged image of V1 expressing Kv-ArcLight-ST shown in (**a**). Average optical traces of fragments of FOV indicated with dotted lines are shown in (**h**). Scale bar, 500 µm. **h** Fluorescence change of Kv-ArcLight-ST. **i** Cross correlation among areas. **j** Relationship between correlation and distance. This result suggests that subthreshold events are roughly correlated among areas. **k** Representative propagation patterns of subthreshold events indicated with red rectangles in (**h**). Event I shows the partial activation in the right side of the FOV, event (II) shows the propagating activity from left to right in the FOV, and event (III) shows nearly synchronous activation in the FOV. (l) Magnified optical traces in areas (I)–(III) indicated with red rectangles in (**h**). Imaging was performed in anaesthetized mice with isoflurane (1.5% v/v). Excitation: 460–500 nm, ~1 mW (~0.05 mW/mm²) at the imaging plane. In each box plots, the central line represents median, and the bottom and the top edges of the box represent 25th and 75th percentile, respectively. The lower and the upper whiskers extend to the minimum and the maximum data points, respectively. Numbers on the graphs represent $P$ values.

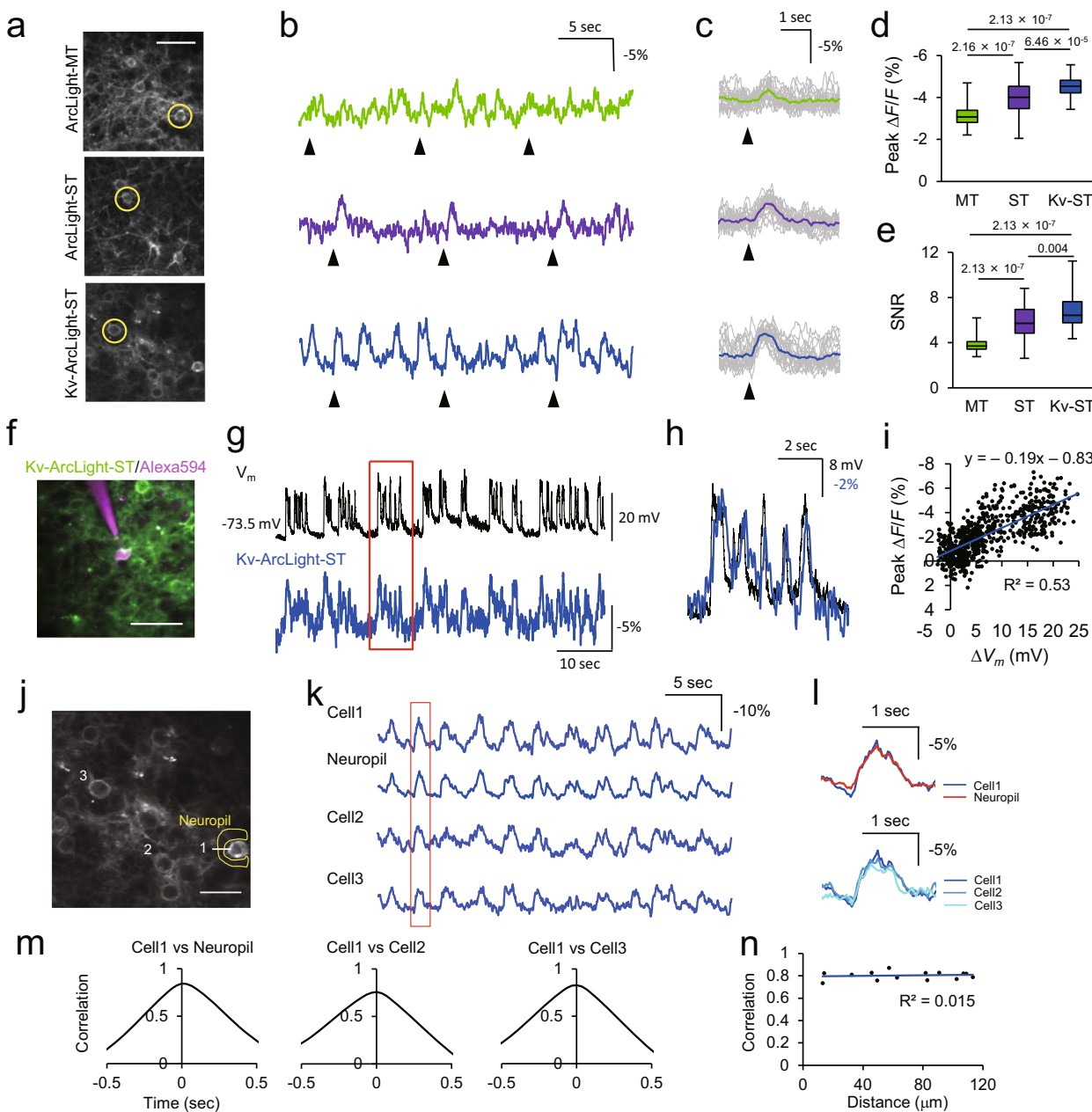

**Fig. 3 Performance of ArcLight variants with two-photon imaging in vivo. a** Two-photon images of cells expressing ArcLight variants in V1. Scale bar, 40 μm. **b** Representative traces of ArcLight variants in the cells indicated with circles in (**a**) in vivo. Arrowheads indicate timing of visual stimulation with flash light of 10 ms. **c** Stimulus-triggered average. Gray traces are individual trials, and colored traces are average of 20 trials. Arrowheads indicate timing of visual stimulation. **d, e** Peak $\Delta F/F$ (**d**) and SNR (**e**) for visually evoked optical signals. $n = 56$ cells from 2 mice (ArcLight-MT), 77 cells from 5 mice (ArcLight-ST) and 62 cells from five mice (Kv-ArcLight-ST). Steel-Dwass test was used for statistical analysis. **f** A two-photon image of a Kv-ArcLight-ST-expressing neuron recorded with whole-cell patch-clamp in vivo. Alexa594 was filled through a patch pipette. Scale bar, 50 μm. **g** Simultaneous current-clamp recording and two-photon voltage imaging in vivo. **h** Magnified optical trace overlaid with electrical trace in the area shown with a red rectangle in (**g**). **i** Correlation between electrical and optical signals (2 cells from 2 mice). **j–n** Spatiotemporal structure of subthreshold events in the local cortical circuits in vivo. **j** Two-photon image of layer 2/3 neurons expressing Kv-ArcLight-ST. Optical traces of cells and a neuropil indicated with white number and yellow characters are shown in (**k**). Scale bar, 25 μm. **k** Fluorescence change of Kv-ArcLight-ST. **l** Magnified traces in the area shown with a red rectangle in (**k**). **m** Cross correlation between cells and neuropil. **n** Relationship between correlation and distance of neurons. Imaging was performed 150–300 μm below the pial surface (two-photon) in lightly anaesthetized mice (isoflurane, ~1.5% v/v). Excitation: 940 nm, 130–150 mW (0.50–0.61 μW/pixel) at the imaging plane. In each box plots, the central line represents median, and the bottom and the top edges of the box represent 25th and 75th percentile, respectively. The lower and the upper whiskers extend to the minimum and the maximum data points, respectively. Numbers on the graphs represent $P$ values.

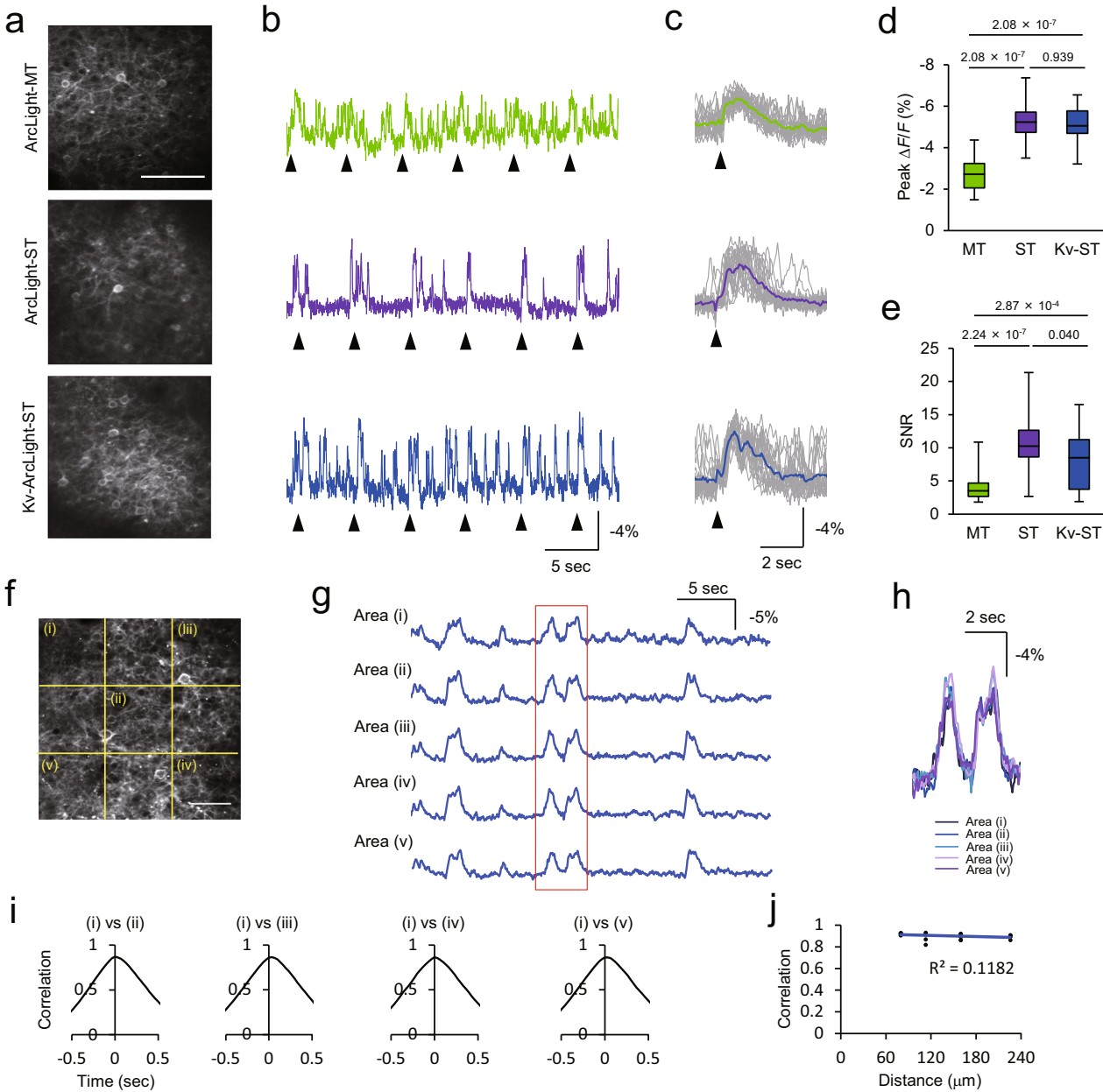

**Fig. 4 Two-photon imaging of optical field potential (OFP) with ArcLight variants in vivo. a** Two-photon image of layer 2/3 neurons expressing ArcLight variants in V1. Average optical traces of entire field-of-view are shown in (**b**). Scale bar, 100 μm. **b, c** Fluorescence change of ArcLight variants (**b**) and stimulus-triggered average over 20 visual stimuli (**c**). Individual trials are shown with gray. Arrowheads indicate the timing of visual stimuli with flash light for 10 ms. **d, e** Peak ΔF/F (**d**) and SNR (**e**) of OFP in response to visual stimuli. Steel-Dwass test was used for statistical analysis. $n = 40$ events for each condition. Visual response was recorded 20 times in each FOV. Two mice were examined in each condition. **f** Two-photon image of primary visual cortex expressing Kv-ArcLight-ST under CAG promoter. Optical traces of small areas indicated with yellow dotted lines are shown in (**g**). Scale bar, 50 μm. **g** Fluorescence change of Kv-ArcLight-ST. **h** Magnified traces in the area shown with a red rectangle in (**g**). Optical signals of neighboring areas showed similar dynamics. **i** Cross-correlation between areas. **j** Relationship between correlation and distance areas. Two-photon imaging was performed in anaesthetized mice with isoflurane (1.5% v/v) in vivo. Excitation: 940 nm, 130–150 mW (0.50–0.61 μW/pixel) at the imaging plane. In each box plot, the central line represents the median, and the bottom and the top edges of the box represent the 25th and 75th percentile, respectively. The lower and the upper whiskers extend to the minimum and the maximum data points, respectively. Numbers on the graphs represent P values.

acute brain slices. Using Kv-ArcLight-ST and the red calcium-sensitive dye Cal-590[16], we successfully discriminated subthreshold and suprathreshold activity (Fig. 5). We next explored in vivo and co-expressed Kv-ArcLight-ST and jRGECO1a in neocortical layer 2/3 pyramidal neurons under the CAG promoter using in utero electroporation (Fig. 6a). Simultaneous two-photon voltage and calcium imaging also discriminated spiking and subthreshold events with cellular resolution under light

anesthesia (isoflurane, ~1.5% v/v) in vivo (Fig. 6b). During these recordings, we applied brief visual stimuli of 10 ms and analyzed the data with a stimulus-triggered average (Fig. 6c–e). Comparing spiking and non-spiking events, we found differences in the input–output relationships across neurons. For example, in Fig. 3, neuron 1 showed delayed visually-evoked spikes after spontaneous subthreshold input, whereas neuron 2 showed fast-firing visually evoked spikes in the absence of previous spontaneous

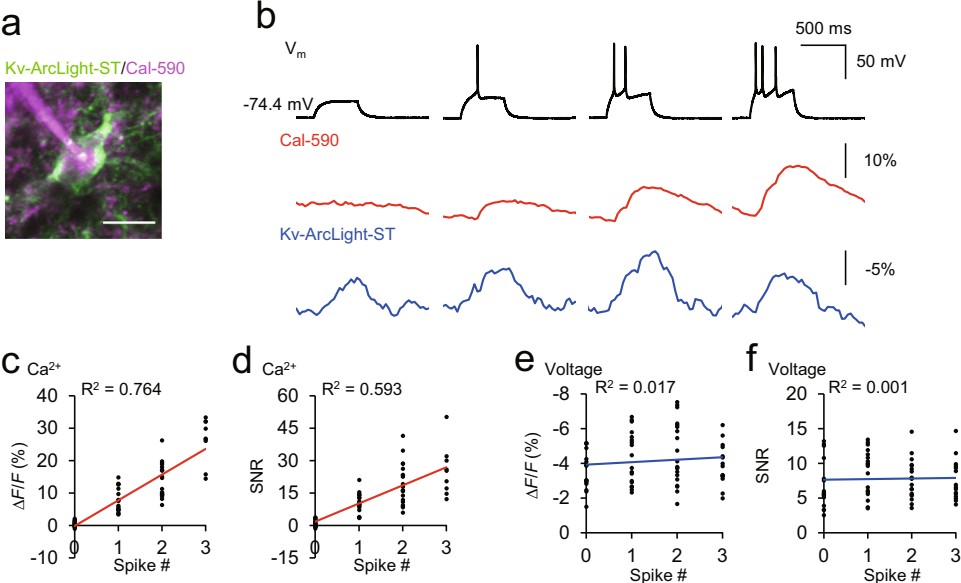

**Fig. 5 Discrimination of spiking and subthreshold events with simultaneous two-photon voltage and calcium imaging in brain slices. a** Two-photon image of layer 2/3 neurons expressing Kv-ArcLight-ST. Red calcium indicator, Cal-590 was filled through a patch pipette. Scale bar, 20 μm. **b** Simultaneous whole-cell recording, two-photon voltage, and calcium imaging. Depolarizing current steps were applied to induce subthreshold depolarization, single and multiple action potentials. Resting potential was -68.5 ± 2.6 mV (mean ± SEM, $n = 8$ cells). **c**, d Correlation between spike number and fluorescence change (**c**) and SNR (**d**) of Cal-590. Cal-590 showed spike number-dependent fluorescence change, and optical signal was not detectable for subthreshold events ($\Delta F/F$, 0.57 ± 0.21%; SNR, 1.10 ± 0.38, mean ± SEM). **e, f** Correlation between spike number and fluorescence change (**e**) and SNR (**f**) of Kv-ArcLight-ST. Optical signal of Kv-ArcLight-ST was not spike number-dependent, but was sensitive to subthreshold events ($\Delta F/F$, −3.41 ± 0.33%; SNR, 6.57 ± 0.91, mean ± SEM). Large depolarizing events ($\Delta V_m > 20$ mV) were analyzed. Eight cells from three mice were examined. Excitation: 990 nm, 150–180 mW (0.57–0.69 μW/pixel) at the imaging plane.

input (Fig. 6e–g). In neuron 3, the timing of visually evoked potentials was faster in spiking events than non-spiking events (Fig. 6e, f, h). These examples demonstrate that the same sensory stimulation can generate different patterns of synaptic inputs in different neurons and that neurons process input signals differently to generate action potentials.

**Divergent pre-ictal dynamics of subthreshold potentials and intracellular calcium in a pharmacological model of spreading focal seizures.** Finally, to test the applicability of the method to a biological question, we imaged acutely induced focal seizures whose precise sub- and suprathreshold neural dynamics during seizure formation and spread[17–20] remain of interest for a deeper understanding of local seizure progression. We locally injected small amounts of 4-aminopyridine (4-AP, see the "Methods" section) into primary visual cortex, and recorded electrical LFP at the 4-AP injection site in lightly anaesthetized mice (isoflurane, ~1.5% v/v). At the same time, we performed two-photon, two-color voltage and calcium imaging at a distance of 1–1.5 mm (Fig. 7j). Demonstrating the robustness of our approach, we successfully recorded LFP, calcium and voltage dynamics simultaneously for a duration of 44 min (Supplementary Fig. 10). We found that, during LFP seizure onset, subthreshold depolarizations showed fast propagation (signal delay between initiation site and imaged area) and were nearly synchronized at the imaging site, whereas population calcium transients propagated slowly, and sequential calcium transients were observed in the neurons at the imaging site (Figs. 7c, d and 8). In the early interictal (period between two seizures) interval, interictal LFP spikes (IIS) were detected electrically at the initiation site, but neither by distant voltage nor calcium imaging (Fig. 7e, f), indicating prevailing surround inhibitory effects, and thus little net-excitatory subthreshold input from the initiation site. Yet, preceding the seizure,

IIS were detected electrically and population EPSPs were detected with voltage imaging, but not with calcium imaging (Fig. 7g, h). During this late interictal period, both LFP and optical voltage signals of Kv-ArcLight-ST increased and became increasingly correlated towards electrographic seizure onset (Fig. 7i–k). Using our combined imaging approach, we demonstrate a preictal escalation of excitatory inputs from the initiation site, and an increasing divergence of sub- and suprathreshold activity in surround networks prior to seizure invasion. Thus, Kv-ArcLight-ST and jRGECO1a co-expression in neuronal populations in vivo enables the study of input and output properties of compartmentalized local epileptic networks.

**Discussion**
In this study, we explored an approach for spatiotemporal mapping of input and output in cortical circuits. To achieve this, we combined a green voltage indicator and a red calcium indicator to detect sub- and suprathreshold events, respectively. Despite recent advances in the development of GEVIs, detection of subthreshold events with two-photon imaging, especially in the intact brain, has remained challenging[11]. By introducing a point mutation, we successfully improved ArcLight-MT for detection of subthreshold membrane potentials in neuronal populations in vivo[11,12]. ArcLight-ST and Kv-ArcLight-ST faithfully detected spontaneous and visually evoked UP states from population of neurons with one-photon wide-field in vivo imaging despite the presence of putative heart beat artifact. Further, ArcLight-ST could report spontaneous and visually evoked depolarization in cellular and subcellular compartments, while Kv-ArcLight-ST was used for multicellular imaging of subthreshold potentials with two-photon imaging in vivo. Importantly, Kv-ArcLight-ST displayed high photostability in our experimental framework, and thus facilitated prolonged imaging sessions (~45 min, potentially

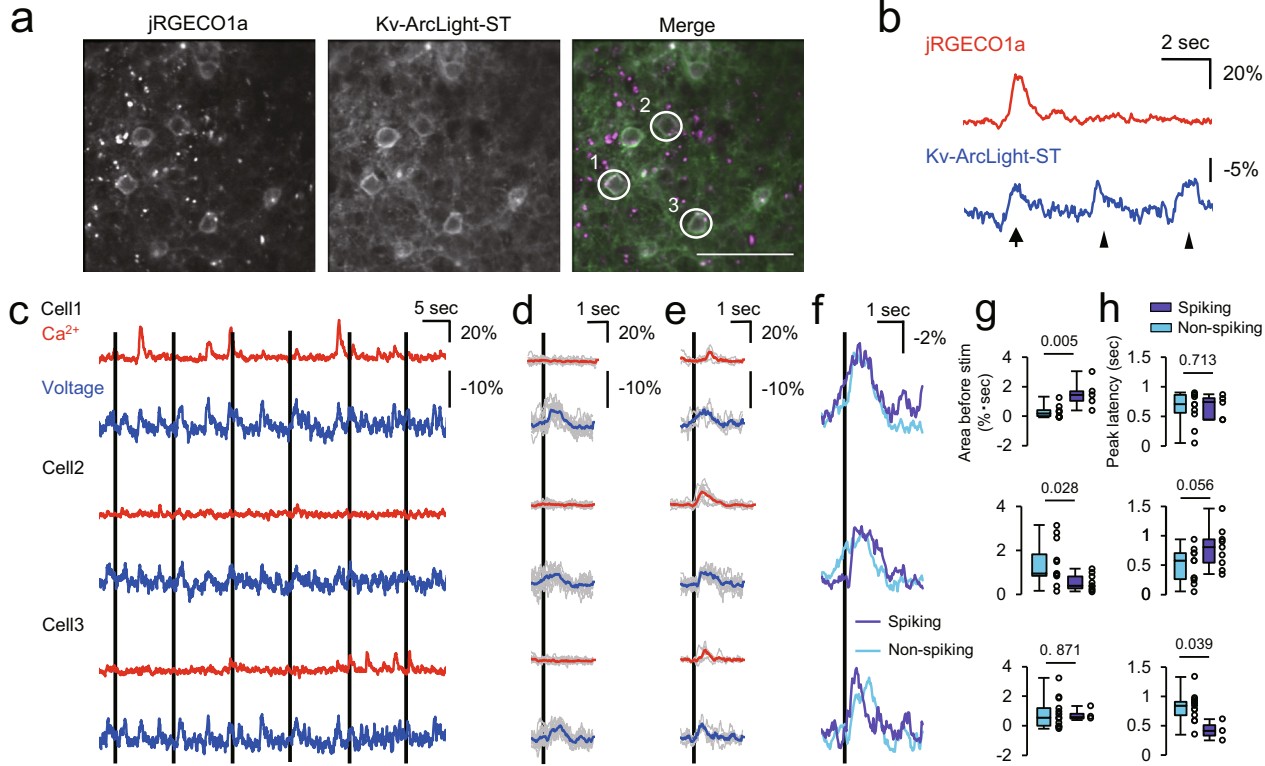

**Fig. 6 Simultaneous two-photon voltage and calcium imaging in physiological conditions in vivo. a** wo-photon images of layer 2/3 pyramidal neurons co-expressing Kv-ArcLight-ST and jRGECO1a in V1 in vivo. Scale bar, 50 μm. **b** Simultaneously recorded dynamics of calcium and voltage. Arrow indicates spiking event, and arrowheads putative subthreshold events. **c** Multi-cell recording of voltage and calcium dynamics. Black lines indicate timing of visual stimulation. Cell numbers correspond with those in (**a**). **d, e** Stimulus-triggered average of non-spiking (**d**) and spiking (**e**) events. Gray traces are individual trials, and colored traces are average traces. Black line indicates the timing of visual stimulation. **f** Average voltage traces for spiking and non-spiking events. Black line indicates timing of visual stimulation of 10 ms. **g, h** Area of membrane potential change for 500 ms ahead of visual stimulation (**g**) and Peak latency of visually evoked potential from the timing of stimulation (**h**). $n = 10$ events (cell 1, non-spiking), 5 (cell 1, spiking), 11 (cell 2, non-spiking), 9 (cell 2, spiking), 13 (cell 3, non-spiking), 3 (cell 3, spiking). Mann–Whitney test was used for statistical analysis. Kv-ArcLight-ST and jRGECO1a were expressed under a CAG promoter using in utero electroporation. Imaging was performed 150–300 μm below the surface (two-photon) in lightly anaesthetized mice (isoflurane, ~1.5% v/v). Excitation: 990 nm, 150–180 mW (0.57–0.69 μW/pixel) at the imaging plane. In each box plots, the central line represents median, and the bottom and the top edges of the box represent 25th and 75th percentile, respectively. The lower and the upper whiskers extend to the minimum and the maximum data points, respectively. Numbers on the graphs represent *P* values.

even longer). While ArcLight-ST and Kv-ArcLight-ST represent useful tools for the neuroimaging community, there are limitations using ArcLight-ST and Kv-ArcLight-ST. First, slow kinetics of ArcLight-ST and Kv-ArcLight-ST make it difficult to detect fast subthreshold oscillations, e.g. in the gamma frequency range. Secondly, the detection limit of subthreshold potentials was ~15 mV with Kv-ArcLight-ST. Thus, ArcLight-ST and Kv-ArcLight-ST primarily detect large and slow voltage changes. The introduction of newer GEVIs could supersede some of these limitations[21].

In the imaged cortical circuits, large and slow subthreshold events were highly correlated among cells and neuropil compartments, and their correlation was not significantly different across distance among pairs of cells or neuropil compartments within the imaged field of view. These results suggest that large and slow subthreshold events are nearly synchronous in local cortical circuits, and are consistent with previous studies performing multiple whole-cell recordings or simultaneous whole-cell and LFP recordings in vivo[1,22]. By contrast, many different spatiotemporal patterns of subthreshold activity were observed with wide-field imaging in the primary visual cortex. Thus, our approach may be particularly helpful for the study of circuit mechanisms and physiological roles of local and global

synchronization, especially in combination with opto- or chemogenetics, and behavioral context.

Although the sensitivity of ArcLight-ST and Kv-ArcLight-ST was improved for subthreshold potentials, they are less sensitive to action potentials. To compensate this disadvantage of ArcLight-ST and Kv-ArcLight-ST, we combined them with established calcium indicators to simultaneously record subthreshold inputs and action potentials from multiple neurons in vivo. Using this dual approach, we first successfully discriminated spiking and subthreshold events in acute brain slices, and observed UP states with and without calcium transients. These results prompted us to perform simultaneous two-photon voltage and calcium imaging of spiking and non-spiking subthreshold events in vivo, where we found a variable input–output relationship among neurons. Thus, while it remains incompletely understood how divergent neural response properties are generated within local circuits, our approach could be helpful in addressing this fundamental issue, particularly when applied in concert with optogenetics and pharmacological experiments. Finally, we found divergent in vivo patterns of sub- and suprathreshold activities during pharmacologically induced focal seizures[19]. During the early interictal period, neither sub- nor suprathreshold activity were observed at a distance to the 4-AP

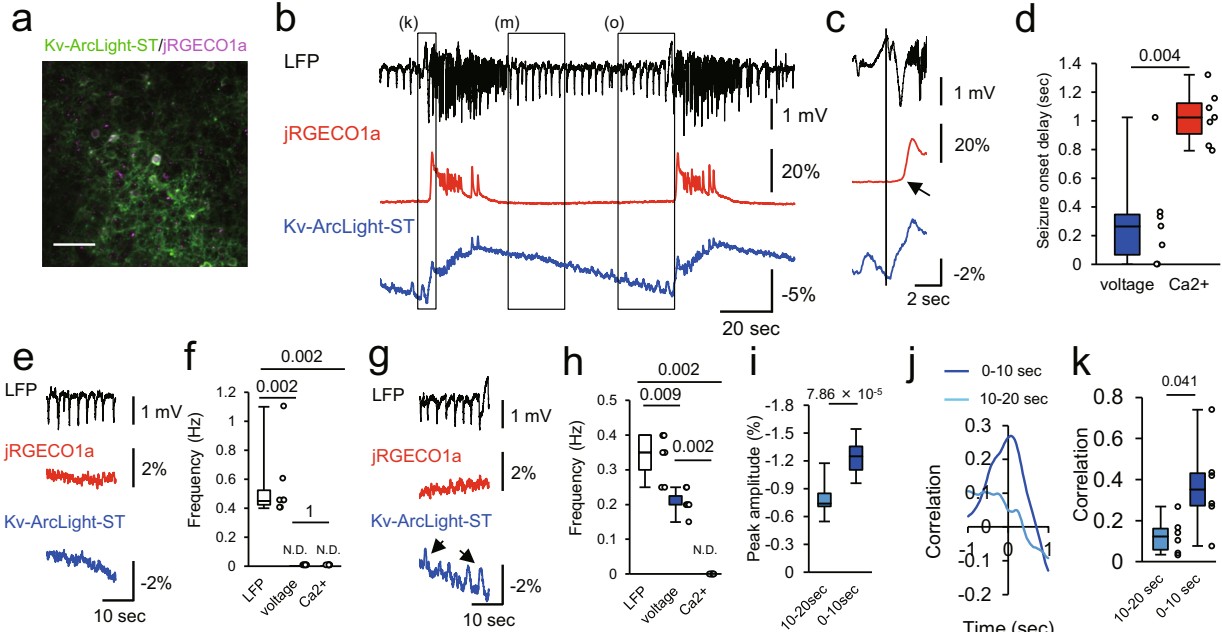

**Fig. 7 Simultaneous two-photon voltage and calcium imaging in pathophysiological conditions in vivo. a** A two-photon image of layer 2/3 pyramidal neurons co-expressing Kv-ArcLight-ST and jRGECO1a in V1. 4-AP was injected about 1–1.5 mm away from the imaging site, while LFP was recorded at the injection site. Scale bar, 50 μm. **b** LFP, calcium and voltage dynamics during epileptic activity. OFP signals were analyzed. **c** Magnified traces around onset of seizure indicated with left rectangle in (**a**). Black line indicates the onset of seizure. Arrow indicates delayed onset of calcium transient. **d** Onset delay of voltage and calcium signals at the imaging site. $n = 6$ events from 2 mice. Mann–Whitney test was used for statistical analysis. **e** Interictal period after seizure indicated with middle rectangle in (**b**). **f** Frequency of interictal events. $n = 6$ events from 2 mice. Steel-Dwass test was used for statistical analysis. N.D., not detected. **g** Interictal period right before seizure onset indicated with rectangle on the right side in (**b**). **h** Frequency of interictal events. $n = 6$ events from 2 mice. Steel-Dwass test was used for statistical analysis. ND not detected. Arrows indicate interictal events detected at the imaging site. **i** Peak amplitude of interictal EPSPs recorded with Kv-ArcLight-ST at 0–10 s before seizure and 10–20 s before seizure onset. $n = 13$ events in each condition. Mann–Whitney test was used for statistical analysis. **j, k** An example of cross-correlation between LFP and signal of Kv-ArcLight-ST at 10–20 s before LFP seizure onset and 0–10 s before seizure during the late interictal period shown in (**g**). $n = 6$ events in each condition. The Steel-Dwass test was used for statistical analysis. In each box plots, the central line represents median, and the bottom and the top edges of the box represent 25th and 75th percentile, respectively. The lower and the upper whiskers extend to the minimum and the maximum data points, respectively. Numbers on the graphs represent $P$ values.

injection site, where interictal spikes were recorded in the LFP. However, towards the late interictal period, LFP interictal spikes started to trigger subthreshold activity in the imaged field of view (FOV), and the two voltage signals became increasingly correlated towards the following seizure. However, no calcium transients were observed in the FOV during this time. In line with previous in vivo work, this suggests pre-ictal synaptic barrages originating from the seizure initiation site while neural firing in surround territories is suppressed by yet intact inhibitory circuits[2,20,23–25]. Upon electrographic seizure onset, the arrival of the slowly traveling ictal wavefront and with it the recruitment of neurons in the FOV was observed by calcium imaging[17,19,24,26,27], whereas subthreshold activity was nearly synchronized with the 4-AP injection site. These data highlight the dynamic relationship between focal sub- and suprathreshold activity during focal seizure emergence and spread, and demonstrate how our approach could be a valuable addition to the portfolio of imaging strategies to address key questions of pathologically transformed neural circuits in vivo.

The slow kinetics and limited resolution of ArcLight for subthreshold potentials (change of fluorescence per mV) still necessitates further development towards fast and sensitive subthreshold voltage indicators. However, given the different technical demands of subthreshold and spike imaging, a one-shoe-fits-all solution to both technical challenges will likely be limiting. Here, we have shown that the combination of different voltage

and calcium imaging sensors could be a powerful approach for meeting this technical challenge, and demonstrate the capability of combinatorial two-photon, two-color voltage and calcium imaging in spatiotemporal mapping of the input–output relationship of neural circuits during physiological and pathophysiological conditions in vivo.

## Methods

**Animals**. Wild type CD1; ICR mice of both sexes (Charles River) were used for all experiments. Animals were maintained in a temperature-controlled environment on a 12 h light–dark cycle with food and water ad libitum. Animal room was kept between 20 and 25 °C, and between 40% and 60% humidity. Experimental procedures were carried out in accordance with the guidelines for animal care and use of the U.S. National Institute of Health, Columbia University and U.S. Army Research Office. All protocols using animals were approved by Institutional Animal Care and Use Committees in Columbia University and Animal Care and Use Review Office in U.S. Army Research office.

**Plasmids**. ArcLight-MT was kindly provided by Dr. Vincent Pieribone (Yale University), and was used in the previous study[11,28]. The cDNA encoding jRGE-CO1a with nuclear export signal was purchased from Addgene (plasmid #61563), and subcloned into pCAG vector containing CMV enhancer, CAG promotor and rabbit globin poly-A sequences. The C-terminal fragment of mouse $K_v2.1$[5,8] was synthesized by Thermo Fisher Scientific (sequence: 5′-TTGGCAAAGAATTCGCC ACCATGCAGTCCCAGCCCATCCTCAACACCAAGGAGATGGCCCCGCA GAGCAAGCCTCCAGAGGAGCTGGAGATGAGCAGCATGCCCAGCCC AGTGGCACCTCTGCCCGCACGCACGGAGGGCGTCATCGACATGCGGAG CATGTCCAGCATTGACAGCTTCATCAGCTGTGCCACGGACTTC CCTGAAGCCACCAGATTCGGTACCGCAATGGAAGGTTT-3′), and was

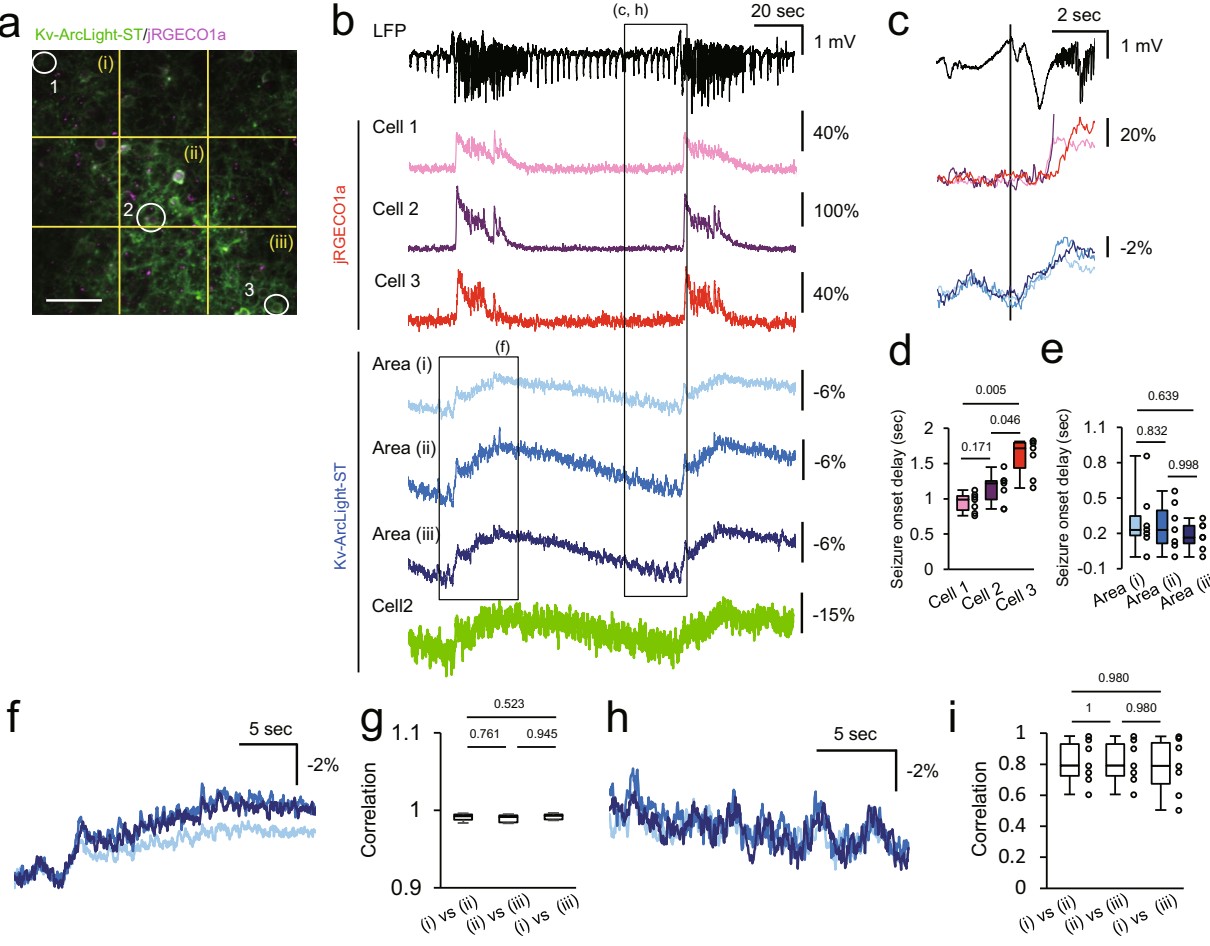

**Fig. 8 Spatiotemporal structure of sub- and suprathreshold events in acute focal epileptic seizures in vivo. a** Two-photon image of layer 2/3 neurons expressing Kv-ArcLight-ST and jRGECO1a under CAG promoter in V1. Cells and neuropil indicated with white numbers and yellow characters; their optical traces are shown in (**b**). Scale bar, 50 μm. **b** LFP at the 4-AP injection site and fluorescence change of Kv-ArcLight-ST and jRGECO1a about 1–1.5 mm away from the injection site. **c** Magnified traces from the dotted rectangle in (**b**). Optical signals of calcium and voltage showed different dynamics at LFP seizure onset. At the imaging site, suprathreshold calcium imaging showed sequential recruitment of neighboring cells, whereas voltage signals were nearly synchronized. **d, e** Suprathreshold (calcium imaging) (**d**) and subthreshold (voltage imaging) during LFP seizure onset (**e**). $n = 7$ events from 2 mice. Steel-Dwass test was used for statistical analysis. **f** Optical signals of Kv-ArcLight-ST in 3 areas during seizure. Colors of traces correspond to those in (**b**). **g** Correlation of OFP in neighboring areas during electrographic (LFP) seizure. $n = 7$ events from 2 mice. **h** Optical signals of Kv-ArcLight-ST in three areas during late interictal period. Individual trace colors correspond to those in (**b**). **i** Correlation of OFP in neighboring areas during late interictal period. Optical signals showed similar dynamics in the local cortical circuit in the imaged field of view. $n = 7$ events from 2 mice. Two-photon imaging was performed in lightly anaesthetized mice with isoflurane (~1.5% v/v). Excitation wavelength was 990 nm, and excitation power was 150–180 mW (0.57–0.69 μW/pixel) at the imaging plane. Steel-Dwass test was used for statistical analysis. In each box plots, the central line represents median, and the bottom and the top edges of the box represent 25th and 75th percentile, respectively. The lower and the upper whiskers extend to the minimum and the maximum data points, respectively. Numbers on the graphs represent $P$ values.

inserted into the N-terminus of ArcLight-MT with In Fusion cloning kit (Takara/Clontech, 639648). Q217E mutation in ArcLight-MT was introduced with polymerase-chain-reaction (PCR), followed by reaction with In Fusion kit. Sequences of custom primers synthesized by Thermo Fisher Scientific are as follows: forward, 5′-ACTCGGAGAGCTGGTGGTCCTCGCT-3′, reverse, 5′-ACCAGCTCTCCGAGTCCGTCTGCTC-3′. ArcLight variants were subcloned into pCAG vector for in utero electroporation and pCMV vector containing CMV promotor, WPRE and rabbit globin poly-A sequences for transfection in vitro.

**Simultaneous voltage imaging and electrophysiology in vitro.** Dissociated primary hippocampal culture was prepared from embryos of CD1 mice at E18. Cells were plated on 12 mm round coverslips coated with poly-ʟ-lysine (BD Biosciences, P4707-50ML) at $1 \times 10^5$ cells per coverslip. Cultures were maintained in Neurobasal medium (Life Technologies, 21103-049) containing 0.5 mM gluta-mine (Sigma-Aldrich, G7513-100ML), and 2% B27 supplement (Life Technologies, 17504044) at 37 °C with 5% $CO_2$. Cells were transfected with ArcLight variants using standard calcium phosphate methods between days-in-vitro (DIV) 9 and 12.

Briefly, endotoxin-free plasmid DNA (2 μg per well), $CaCl_2$ (final: 250 mM) and HEPES-buffered saline (final: 1 ×, pH 7.05) were mixed with vortex, and incubated for 20 min at room temperature. Five minutes before transfection, neurobasal medium was replaced by 1 ml minimal essential medium (MEM) (Life Technologies, 11090073). The DNA/$CaPO_4$ mixture was added to cells, and cells were incubated for 30 min at 37 °C. The cells were washed with MEM, and MEM was replaced with Neurobasal medium.

Recording was performed 1–2 days after transfection. HEPES-buffered artificial cerebrospinal fluid (ACSF) was used as an external solution (in mM: 145 NaCl, 2.5 KCl, 10 HEPES, 2 $CaCl_2$, 1 $MgCl_2$, 10 glucose, pH 7.3). Patch pipettes of 4–6 MΩ were pulled with a horizontal puller (DMZ-Universal puller, Zeitz Instruments). Patch pipettes were filled with K-Gluconate internal solution (in mM: 120 K-gluconate, 3 KCl, 7 NaCl, 4 Mg-ATP, 0.3 Na-GTP, 20 HEPES, and 14 Tris-phosphocreatine, pH 7.3), and whole-cell recording was performed. Data on electrophysiology was obtained at 10 kHz, and low-pass filtered at 4 kHz using Axon Multiclamp 700B amplifier (Molecular Devices) and custom-made software, PackIO[29] (https://github.com/apacker83/PackIO). Data was discarded from analysis if resting potential was greater than −50 mV or input resistance was smaller than 100 MΩ.

Voltage imaging was conducted with a microscope (BX50WI, Olympus), ×60/1.1 N.A. water-immersion objective (Olympus). Mercury lamp (Osram) was used for excitation of ArcLight variants. Filters were used as follows: a 480/40 excitation filter (Chroma), a dichroic mirror 495lp (Chroma), and a 535/50 m emission filter (Chroma). Excitation power was ~5.5 mW (~36 mW/mm$^2$) at the stage. Images were acquired at 1 kHz using sCMOS camera (Orca-Flash 4.0, Hamamatsu) and HC Image software (Hamamatsu). Data was collected from 2–3 different culture preparations.

**Photobleaching analysis**. To measure photostability with one-photon excitation, cultured neurons expressing ArcLight variants were continuously illuminated with a mercury arc lamp (Osram) in the imaging condition as described above. Images were acquired for 3 min at 10 Hz without patch-clamp recording. For analysis of two-photon photostability, transfected neurons were continuously scanned for 3 min at 30 Hz with Ti:sapphire laser (Coherent) tuned at 940 nm. Excitation power was ~ 5.5 mW (~36 mW/mm$^2$) (1-photon) or 28 mW (~0.11 μW/pixel) (2-photon) at the stage.

**In utero electroporation**. Pregnant CD1 mice at embryonic day 16 (E16) were anaesthetized with isoflurane (3% v/v for induction, 2% v/v for surgery), and their uteruses were exposed. Plasmid solution was prepared at 1 μg/μl (pCAG-ArcLight-MT, pCAG-ArcLight-ST and pCAG-Kv-ArcLight-ST), or 2 μg/μl (pCAG-jRGE-CO1a) in PBS (final: 1x, pH 7.3) with 0.05% Fast Green (Sigma-Aldrich, F7258), and 1–2 μl of DNA solution was injected into the left lateral ventricle using glass micropipettes. Next, three poring electrical pulses (50 V, 10 ms duration, 50 ms interval, 10% decay) followed by three transferring pulses (15 V, 50 ms duration, 50 ms interval, 20% decay) were applied using an electroporator (NEPA21, Nepa Gene), tweezer-type, round, platinum electrode (CUY650P5, Nepa Gene) and a round, platinum electrode (CUY700P4L, Nepa Gene).

**Animal surgery for in vivo recording**. Electroporated mice between postnatal day 30 (P30) and P60 were used for experiment. Mice were anaesthetized with iso-flurane (3.0% v/v for induction, 2.0% for recording). A custom-made titanium head plate was fixed to the skull with dental cement (Parkell). Cranial window of 2 mm$^2$ (for two-photon imaging), 1.5 mm$^2$ (for whole-cell recording) or 3 mm$^2$ (for one-photon wide-field imaging) was made on the left primary visual cortex (2.5 mm lateral from the lambda). The coverslip was placed and sealed on the craniotomy with a cyanoacrylate adhesive. For experiments involving acute focal seizures, a small opening was maintained at the edge of the cranial window to allow insertion of a glass pipette for local 4-AP injection into the cortex (layer 5).

**One-photon wide-field voltage imaging in vivo**. One-photon wide-field voltage imaging was performed with a BX61WI microscope (Olympus) equipped with ×4/0.28 NA dry objective lens (Olympus) and sCMOS camera (Orca-Flash 4.0, Hamamatsu). 470 nm fiber-coupled LED (M470F3, Thorlabs) and a dichroic mirror (495lp, Chroma) was used for excitation of ArcLight variants. Fluorescence was isolated with an emission filter (535/50 m, Chroma). During imaging, mice were head-fixed, and anaesthetized with isoflurane (1.5–2.0% v/v). Images of 128 × 128 pixels (8 × 8 binning) were acquired at 30 Hz. Visual stimuli were applied to contralateral eye using a white LED flash light of 10 ms. Visual stimulation was controlled with Mater-8 plus pulse generator (A.M.P.I.). Excitation power was ~1 mW (~0.05 mW/mm$^2$) at the stage.

**In vivo two-photon imaging**. Mice were anaesthetized with isoflurane (1.5% for recording). Two-photon imaging in vivo was performed with a Prairie microscope (Bruker), ×25/1.05 NA water-immersion objective lens (Olympus) and tunable Ti-sapphire laser (Coherent Chameleon Ultra II, Coherence). Voltage and calcium dynamics were recorded from layer 2/3 pyramidal neurons. The laser was tuned at 940 or 990 nm for imaging of ArcLight variants and 990 nm for imaging of jRGECO1a and dual voltage/calcium imaging. Fluorescent signals were split with HQ 525/70m-2p (for ArcLight signal, Chroma Technology), HQ 607/45-2p (for jRGECO1a signal, Chroma Technology), and 575LP dichroic mirror. Images were acquired at 30 Hz, 512 × 512 pixels with 2× (Figs. 4, 7, 8, Supplementary Figs. 8f–j, 10), 4× (Figs. 3, 5, 6, Supplementary Figs. 5, 6i–n, 7, 8a–e, 9) zoom for 3 min. All experiments were performed with resonant scanning mode. Excitation power was 130–160 mW (0.50–0.61 μW/pixel) (940 nm) or 150–180 mW (0.57–0.69 μW/pixel) (990 nm) at the imaging plane. Visual stimuli were applied to contralateral eye using a white LED flash light of 10 ms. Imaging was performed at 150–300 μm deep from the surface.

**In vivo electrophysiology**. In vivo whole-cell recording was performed based on previous reports[30,31]. In detail, mice were anaesthetized with isoflurane (1.5–2.0% v/v) during the experiment. A square cranial window of 1.5 mm was made, and the dura matter was carefully removed at the access point of the recording pipette. To reduce movement artifacts, the cranial window was covered with 2.0% agarose gel in HEPES-buffered ACSF for in vivo recording (in mM: 150 NaCl, 2.5 KCl, 10 HEPES, 2 CaCl$_2$, 1 MgCl$_2$, pH 7.3). For all experiments, patch pipettes of 6–8 MΩ were used. All experiments were performed at left primary visual cortex (2.5 mm lateral from the lambda), and data was obtained at 10 kHz using Multiclamp 700B amplifier.

For whole-cell recording, patch pipettes were filled with K-gluconate internal solution containing Alexa594 (in mM: 120 K-gluconate, 3 KCl, 7 NaCl, 4 Mg-ATP, 0.3 Na-GTP, 20 HEPES, and 14 Tris–phosphocreatine, 0.025 Alexa594, pH 7.3). High positive pressure (200–250 mbar) was applied to the patch pipettes until they were inserted into the brain, and then pressure was reduced to 30–40 mbar to approach a cell. Access resistance was 20–50 MΩ. Flash light-induced visual response was recorded with current-clamp mode. The data was low-pass-filtered at 4 kHz.

For loose-seal cell-attached recording, patch pipettes were filled with HEPES-buffered ACSF containing 25 μM Alexa488. The pipettes were guided to neurons expressing voltage indicators as described above. Spontaneous action potentials were recorded using voltage-clamp mode at 0 mV together with voltage imaging. The data was low-pass-filtered at 4 kHz.

For recording of the local field potential (LFP) during pharmacologically induced seizures, a glass pipette was filled with HEPES-buffered ACSF containing 4-aminopyridine (4-AP) (15 mM). To induce epileptic activity, a small amount of 4-AP was injected into the cortex with positive pressure (10 psi) using a Picospritzer III system (Parker, Hollis, NH) for the brief duration of 3 seconds[17]. LFP signals were amplified using a Multiclamp 700B amplifier (Axon Instruments), low-pass filtered (300 Hz), and digitized at 1 kHz. For data analysis, LFP signals were band-pass filtered between 0.2 and 50 Hz.

**Simultaneous whole-cell recording and two-photon voltage and calcium imaging in acute brain slices**. Electroporated mice of P11–P14 were deeply anaesthetized with isoflurane (5% v/v), and were sacrificed. Brain was removed, and chilled with ice-cold ACSF (in mM: 124 NaCl, 1.8 KCl, 2.5 CaCl$_2$, 1.3 MgCl$_2$, 1.24 NaH$_2$PO$_4$, 26 NaHCO$_3$, 10 glucose) saturated with 95% O$_2$, 5% CO$_2$. Coronal brain slices of 300 μm thickness were cut in ice-cold slicing solution (in mM: 130 NaCl, 4.5 KCl, 2 CaCl$_2$, 33 glucose, 5 HEPES, pH 7.3) using a vibratome (VT1200S, Leica). Slices were recovered in oxygenated ACSF for longer than 1 h at room temperature. Oxygenated ACSF was used for extracellular solution, and K-gluconate internal solution containing Cal-590 (AAT Bioquest) was used or internal solution (in mM: 112 K-gluconate, 8 KCl, 4 Mg-ATP, 0.375 Na-GTP, 10 HEPES, and 10 Na-phosphocreatine, 0.05 Cal-590, pH 7.3). Glass pipettes of 5–7 MΩ were used for recording. Recording was started 1 min after making whole-cell condition to fill Cal-590 into the cell. Access resistance was 20–45 MΩ. Images of 512 × 512 pixels were acquired at 30 Hz with 4× zoom. The excitation wavelength was 990 nm, and excitation power was 150–180 mW (0.57–0.69 μW/pixel).

**Data analysis, statistics, and reproducibility**. Images were analyzed using ImageJ (NIH) and MATLAB (MathWorks). For in vitro imaging, background was subtracted with ImageJ, and bleach correction was performed with spline function, and $\Delta F/F$ was calculated with MATLAB routine. Region-of-interest (ROI) was manually selected around somata and dendrite. SNR was calculated as peak $\Delta F/F$ over standard deviation of the baseline fluctuation for 50 ms before application of voltage steps or current injection. In voltage-clamp experiments, the rise and decay time constant were determined by double-exponential fit. In current-clamp experiments for spike detection, rise time was defined as the time from the starting time of stimulation to the time of peak fluorescence amplitude. Decay time was calculated by fitting to single exponential curve. For photostability analysis, fluorescence was normalized at $t = 0$, and averaged over all cells. Decay curves were fit to a single exponential curve, and time constant was calculated.

To analyze data on one-photon wide-field imaging in vivo, images containing artifact of flash light were replaced by the previous frame. Then, mean fluorescence in the whole field-of-view was calculated. Slow fluctuation of fluorescence in the ROI was corrected by subtracting slow trendline calculated with spline function of MATLAB. Then, $\Delta F/F$ was defined as $(F-F_0)/F_0$, where $F$ was intensity of fluorescence and $F_0$ was the average fluorescence for 0.5 s before stimulation. SNR was calculated as peak $\Delta F/F$ over standard deviation of the baseline fluctuation for 0.5 s before visual stimulation.

For analysis of in vivo two-photon imaging data, images with artifact of flash light were replaced by the previous frame. Movement was corrected using TurboReg plugin, and images were filtered with Kalman stack filter using ImageJ. Acquisition noise variance estimation was set at 0.05, and bias to be placed on the prediction was set at 0.80. Then, ROI was manually set, and mean fluorescence in the ROI was calculated. Slow fluctuation of fluorescence in the ROI was corrected by subtracting slow trendline calculated with spline function of MATLAB. Then, $\Delta F/F$ was defined as $(F-F_0)/F_0$, where $F$ was intensity of fluorescence and $F_0$ was the average fluorescence for 0.5 s before stimulation. SNR was calculated as peak $\Delta F/F$ over standard deviation of the baseline fluctuation for 0.5 s before visual stimulation. $R^2$ between electrical and optical signals in vivo was calculated as $R^2 = 1-$(total sum of squares)/(residual sum of square) using MATLAB.

Statistical analysis (two-sided Mann–Whitney test or Steel-Dwass test) was performed with KyPlot5.0 (KyensLab). $P$ values < 0.05 was considered as significant. The number of samples are described in figure legends. The data was reproduced from two different culture preparations (Fig. 1b, d, f, Supplementary Figs. 1c, 2b, c, f, g, 3b, d, e, 4b, d, 5c, d, k–n), 5 mice (Fig. 2e, f, i, j, Supplementary Fig. 5g, h), 2 mice (ArcLight-MT, Fig. 3d, e), 5 mice (ArcLight-ST, Fig. 3d, e), 3

mice (Kv-ArcLight-ST, Fig. 3d, e), 2 mice (Figs. 3i, m, n, 4d, e, i, j, 6g, h, 7d, f, h, i–k, d, e, g, i, Supplementary Figs. 6d, e, 7d–g, Supplementary Fig. 9c), 3 mice (Fig. 5c–f, Supplementary Fig. 8d, e, i, j).

**Reporting summary**. Further information on research design is available in the Nature Research Reporting Summary linked to this article.

## Data availability

Source data are provided with this paper. Other data that support the findings of this study are available upon reasonable request because of too large data size to share in the public depository.

## Code availability

Analysis codes were previously published[11], and are available in Mendeley (https://doi.org/10.17632/8rxrc428bp.2).

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

## Acknowledgements

We thank Drs. Albert Lee and Shinsuke Tanaka (Janelia Research Campus) for teaching the whole-cell patch-clamp experiment in vivo. We also thank Reka Letso, Mari Bando, and Alexa Semonche for preparing primary hippocampal cultures, members of the Yuste laboratory and Dr. Naoki Honkura (Hamamatsu University of School of Medicine) for their helpful comments. This work was supported by the National Eye Institute (NEI) (R01EY011787), National Institute of Neurological Disease and Stroke (R01NS110422; R34NS116740), and the National Institute of Mental Health (NIMH) (R01MH100561). This material is based on work fully or partly supported by the US Army Research Laboratory and the US Army Research Office (ARO) under contract number W911NF-12-1-0594 (MURI). Y.B. was supported by The Uehara Memorial Foundation, The Japan Society for the Promotion of Science (JSPS) (20K15914), The Japan Science Society (Sasakawa Scientific Research Grant, 2019-4055), Hamamatsu Foundation for Science and Technology Promotion and Takeda Science Foundation. M.W. was supported by Deutsche Forschungsgemeinschaft (DFG, Grant WE 5517/1-1). M.W. is a fellow within the Hertie Network of Excellence in Clinical Neuroscience.

## Author contributions

Y.B. and R.Y. designed the project. Y.B and M.W. performed experiments, and analyzed data. Y.B., M.W., and R.Y wrote the manuscript. R.Y. assembled and directed the team and provided resources and equipment.

## Competing interests

The authors declare no competing interests.
