## [Peer Review File · Nature Communications]

Simultaneous two-photon imaging of action potentials and subthreshold inputs in vivoEditorial Note: This manuscript has been previously reviewed at another journal that is not operating a transparent peer review scheme. This document only contains reviewer comments and rebuttal letters for versions considered at *Nature Communications*.

REVIEWER COMMENTS

Reviewer #1 (Remarks to the Author):

Although the manuscript is improved, several issues remain with Supplementary Fig. 7:

We requested a comparison of soma-targeted ArcLight-ST with soma-targeted ASAP3 (since soma-targeted ASAP3 is what was previously used in vivo), but the authors instead compared the non-soma-targeted indicators in the new Supplementary Fig. 7. I guess this is fine, but not ideal.

Supplementary Fig. 7 – the bottom panels of a and c do not have scale bars

Supplementary Fig. 7 – the density of labeling looks much higher for ASAP3 compared with ArcLight-ST, which would likely affect the SNR calculations in favor of ArcLight-ST. The authors should show images of all of the fields-of-view imaged for this figure so that the reader can judge if the comparison was fair.

Supplementary Fig. 7 – Most of the figure caption still refers to “Kv-ArcLight-ST” when it should be “ArcLight-ST”.

Supplementary Fig. 7 – It is unclear to me why the $\Delta F/F$ quantifications in panels d and f show that ArcLight-ST and ASAP3 responses were not significantly different when the averaged fluorescence traces in panel c show a clear stimulus-locked response for ArcLight-ST and no obvious response for ASAP3.

Reviewer #2 (Remarks to the Author):

The authors have addressed the concerns of the Reviewers from the previous iterations. Congratulations on the nice work!

Reviewer #1 (Remarks to the Author):

Although the manuscript is improved, several issues remain with Supplementary Fig. 7:

We requested a comparison of soma-targeted ArcLight-ST with soma-targeted ASAP3 (since soma-targeted ASAP3 is what was previously used in vivo), but the authors instead compared the non-soma-targeted indicators in the new Supplementary Fig. 7. I guess this is fine, but not ideal.

We are glad that the reviewer agrees that the comparison between now similarly localized indicators in the revised version is fair. Soma-targeted ASAP3 is most useful to monitor action potentials in the soma, while ArcLight-ST may be used primarily to monitor subthreshold potentials in both soma and dendrite *in vivo*. Thus, we compared non-soma-targeted ArcLight-ST and ASAP3. We believe that we have gone great lengths to demonstrate the usefulness of our ArcLight-ST constructs under physiological and pathological conditions, which the reviewer already mentioned themselves in the previous round of reviews. Across 2-p imaging systems, differences in performance of indicator to previously reported values are not unexpected, and it was not the primary purpose of our paper to compare performance of voltage indicator across several types of expression. We have now provided a fair comparison for the most suitable application of our ArcLight-ST vs. ASAP3, and believe that further comparison of indicators should be up to those optimizing their individual experimental question and framework.

Supplementary Fig. 7 – the bottom panels of a and c do not have scale bars

We apologize for the mistake, and thank the reviewer for making us aware of this. We have added scale bars in the bottom panels of a and c.

Supplementary Fig. 7 – the density of labeling looks much higher for ASAP3 compared with ArcLight-ST, which would likely affect the SNR calculations in favor of ArcLight-ST. The authors should show images of all of the fields-of-view imaged for this figure so that the reader can judge if the comparison was fair.

We thank the reviewer for the helpful comment. Images of entire field-of-view are shown in panel a. We have analyzed both low and high labeled data, and did not observe a difference in SNR with our experimental setup. We have replaced the data of ASAP3 to lower labeled one.

Supplementary Fig. 7 – Most of the figure caption still refers to “Kv-ArcLight-ST” when it should be “ArcLight-ST”.

We apologize for the mistake, and thank the reviewer for pointing this out. We have corrected the caption of Supplementary Fig. 7.

Supplementary Fig. 7 – It is unclear to me why the deltaF/F quantifications in panels d and f show that ArcLight-ST and ASAP3 responses were not significantly different when the averaged fluorescence traces in panel c show a clear stimulus-locked response for ArcLight-ST and no obvious response for ASAP3.

We thank the reviewer for the helpful comment. In our hands, ASAP3 showed a pronounced fluctuation of baseline fluorescence. Similarly substantial fluctuations of baseline fluorescence were also observed in other ASAP variants (ASAP1, 2f and 2s) (Bando et al., Cell Rep., 2019). We think that large $\Delta F/F$ of was observed because of its large baseline fluctuation.

Reviewer #2 (Remarks to the Author):

The authors have addressed the concerns of the Reviewers from the previous iterations. Congratulations on the nice work!

We thank the reviewer for his or her positive comment to our work.